# The mechanism of RNA duplex recognition and unwinding by DEAD-box helicase DDX3X

He Song[1] & Xinhua Ji [1]

DEAD-box helicases (DDXs) regulate RNA processing and metabolism by unwinding short double-stranded (ds) RNAs. Sharing a helicase core composed of two RecA-like domains (D1D2), DDXs function in an ATP-dependent, non-processive manner. As an attractive target for cancer and AIDS treatment, DDX3X and its orthologs are extensively studied, yielding a wealth of biochemical and biophysical data, including structures of apo-D1D2 and post-unwound D1D2:single-stranded RNA complex, and the structure of a D2:dsRNA complex that is thought to represent a pre-unwound state. However, the structure of a pre-unwound D1D2: dsRNA complex remains elusive, and thus, the mechanism of DDX action is not fully understood. Here, we describe the structure of a D1D2 core in complex with a 23-base pair dsRNA at pre-unwound state, revealing that two DDXs recognize a 2-turn dsRNA, each DDX mainly recognizes a single RNA strand, and conformational changes induced by ATP binding unwinds the RNA duplex in a cooperative manner.

[1] Macromolecular Crystallography Laboratory, National Cancer Institute, Frederick, MD 21702, USA. Correspondence and requests for materials should be addressed to X.J. (email: jix@mail.nih.gov)

Found in all kingdoms of life, DEAD-box helicase (DDX) is the largest family of RNA helicases that regulate RNA biogenesis by unwinding short RNA duplexes[1]. Human DDX3X and its yeast ortholog Ded1p represent a subfamily of DDX, which is closely related to the subfamily represented by human DDX4 and fly Vasa (Supplementary Fig. 1)[2]. Like all DDXs, DDX3X contains a helicase core composed of two RecA-like domains (D1D2), harboring 12 highly conserved sequence motifs (Fig. 1a)[3,4]. The D1D2 core of DDX3X is flanked by N- or C-terminal tails, which are largely unstructured but contain motifs responsible for the unique functions of the subfamily[2]. For example, the N-terminal tail of DDX3X contains a nuclear export sequence[5], whereas the C-terminal tail contains a low complexity region that is essential for oligomerization[2,6]. The DDX3X/Ded1p subfamily has generated great interest because every member performs one or more nuclear functions and plays roles in the regulation of translation[2,7]. Recently, the biomedical significance of DDX3X has been rapidly rising because it is closely related to cancer development and progression. Not surprisingly, inhibitors of DDX3X have been developed for cancer treatment[8–10]. Moreover, DDX3X is directly targeted by multiple pathogenic

viruses including HIV[11]. Hence, the enzyme has also been studied as a potential target for the development of anti-AIDS agents.

DDX recognizes short double-stranded RNA (dsRNA) to form a DDX:dsRNA complex; ATP binding to the DDX:dsRNA complex unwinds the RNA duplex, resulting in a complex composed of DDX, single-stranded RNA (ssRNA), and ATP (DDX:ssRNA:ATP). ATP hydrolysis facilitates the release of ssRNA[2–4,12,13]. As such, the functional cycle of DDX can be described by at least three distinct states: the apo-DDX, the pre-unwound DDX:dsRNA assembly, and the post-unwound DDX:ssRNA:ATP assembly. For the apo state, a crystal structure was previously reported for the D1D2 core of DDX3X [Protein Data Bank (PDB): 5E7I][14]. For the post-unwound state, structures were previously determined for several DDXs[15–19], among which the fly Vasa (PDB: 2DB3)[19] is one of the closely related orthologs of DDX3X (Supplementary Fig. 1)[2]. For the pre-unwound state, structure of the Mss116p D2 in complex with a 14-bp dsRNA was previously determined (PDB: 4DB2)[20], but the structure of an intact D1D2 core at the pre-unwound state remains elusive.

The minimal, subfamily-specific functional core of DDX3X was recently redefined to contain not only D1D2 but also short

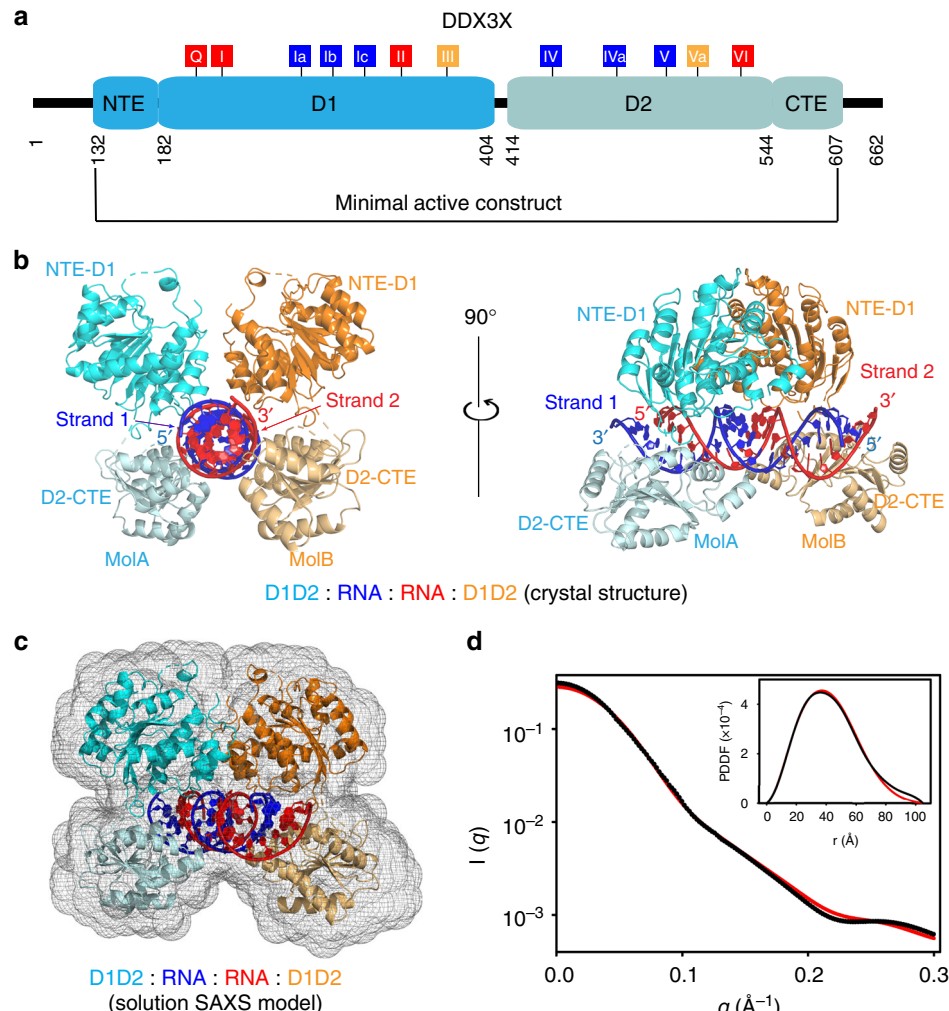

**Fig. 1** Overall structure of the D1D2:dsRNA:D1D2 assembly of human DDX3X. **a** Domain structure of human DDX3X and conserved sequence motifs of DDXs. The minimal active construct of DDX3X helicase core (the D1D2 core, residues 132–607) is used for structural studies. The 12 highly conserved DDX sequence motifs are color coded: red, ATP binding; blue, RNA binding; and orange, coordination between ATP and RNA binding. **b** Crystal structure of the D1D2:dsRNA:D1D2 assembly in two views. The complex contains two D1D2 cores (cyan/orange) and a 23-bp double-stranded RNA (dsRNA) (blue/red). **c** The D1D2:dsRNA:D1D2 structure fits well in the SAXS ab initio envelope. **d** Overlay of experimental scattering profiles (black) with back-calculated scattering profiles (red) for the D1D2:dsRNA:D1D2 complex. The inset shows the overlay of the crystal structure (red) with experimental (black) pair-distance distribution functions (PDDFs). Source data are provided as a Source Data file

N- and C-terminal extensions beyond the two RecA-like domains (NTE and CTE, respectively; Fig. 1a)[14]. Accordingly, the helicase core of DDX3X is the NTE-D1D2-CTE fragment (residues 132–607, Fig. 1a). For the ease of discussion, we refer to the NTE-D1D2-CTE fragment also as the D1D2 core. Here, we report the crystal structure of such a D1D2 core in complex with a 23-bp dsRNA at the pre-unwound state. Our structure reveals that two DDXs recognize a 2-turn RNA duplex and each DDX mainly recognizes one RNA strand. It is known that strand separation occurs as soon as ATP is bound to the pre-unwound DDX: dsRNA complex[3,12,21]. Our structure indicates that the conformational changes induced by ATP binding unwind the RNA duplex in a cooperative manner. It is also known that both DDX3X and Ded1p unwind dsRNA in a cooperative manner[6,22]. In this study, we further performed the Hill cooperativity analysis of DDX3X using two types of biochemical assays, shedding light on the cooperative mechanism of RNA duplex unwinding by DDXs.

## Results

**The crystal structure reveals a D1D2:dsRNA:D1D2 assembly.** Since unwinding occurs once ATP is bound to the DDX:dsRNA complex[3,12,21], we performed crystallization trials using the minimal, subfamily-specific functional core of DDX3X (residues 132–607, Fig. 1a), in the absence of ATP and AMPPNP (a non-hydrolyzable ATP analog), but in the presence of a 26-bp dsRNA with a 2-nucleotide (nt) 3′ overhang on both terminal ends (Supplementary Fig. 2). We obtained crystals of a protein–RNA assembly and determined the structure at 2.5-Å resolution (Table 1).

Our structure shows that the assembly is composed of two D1D2 cores and one RNA duplex (D1D2:dsRNA:D1D2, Fig. 1b). Crystal symmetry is $P3_1$ and the asymmetric unit contains two polypeptide chains (amino acid residues 134–580), two RNA

strands (nucleotide residues 1–23 and 4–27, respectively), seven chloride ions, and 72 water oxygens. Electron density is absent for a total of 32 amino acid residues in each D1D2 core, mainly located on flexible loops, indicating that they are disordered. On one terminal end of the dsRNA, three pairs of nucleotide residues together with the 2-nt 3′ overhang are disordered, whereas on the other terminal end of the dsRNA, the last nucleotide residue of the overhang is disordered. As such, the observed RNA contains 23 bp with a 1-nt 3′ overhang on one terminal end of the duplex. It is known that most DDXs do not have inherent substrate specificity[1]. Our structure shows that the two D1D2 cores jointly recognize base pairs 1–22 of the 26-bp dsRNA so that the resulting complex is not symmetric (Supplementary Fig. 2a). If the two cores recognize base pairs 3–24 of the 26-bp dsRNA (Supplementary Fig. 2b), the crystal could exhibit higher symmetry with one polypeptide chain and one RNA strand in the asymmetric unit.

Our structure of the D1D2:dsRNA:D1D2 complex resembles a clamp that cradles the dsRNA in the middle of four RecA-like domains (two D1s and two D2s). In Fig. 1b, the view on the left is along the axis of dsRNA, showing the tight fit of dsRNA to the two D1D2 cores, whereas the view on the right is perpendicular to the axis of dsRNA, showing that the two D2s span the entire length of the 2-turn, A-form dsRNA, and the two D1s reside on the opposite side of dsRNA.

**The D1D2:dsRNA:D1D2 assembly is also observed in solution.** We further elucidated the formation of the complex in solution by small-angle X-ray scattering (SAXS). Despite the resolution limit of SAXS, this technique provides reliable structural information in solution when it is used in conjunction with crystal structure information[23,24]. We found that the D1D2:dsRNA: D1D2 assembly observed in solution is equivalent in its composition and conformation to the crystal structure (Fig. 1c). The experimental scattering profiles, with the scattering intensity $I(q)$ plotted vs. momentum transfer $q$, along with pair-distance distribution function (PDDF) are shown (Fig. 1d). Analysis of the SAXS data revealed a molecular mass of 120 kDa, in agreement with that of the D1D2:dsRNA:D1D2 complex seen in crystal lattice (124 kDa). The $D_{max}$ value (105 ± 5 Å) is also consistent with the longest dimension of the crystal structure (99 Å).

To illustrate the solution structure of the D1D2:dsRNA:D1D2 assembly, an ab initio shape envelope was built using the program DAMMIN[25]. As shown, the crystal structure fits very well into the ab initio shape envelope (Fig. 1c). The back-calculated scattering profile for the crystal structure also agrees well with the experiment data (Fig. 1d, $\chi^2 = 1.72$). In addition, the predicted $R_g$ value based on the crystal structure (33.4 Å) is virtually identical to that derived from the SAXS data (33.8 ± 0.3 Å). Therefore, we conclude that our crystal structure represents the pre-unwound assembly of DDX3X, providing urgently needed insights into the mechanism of DDX action.

**Integrity of the core is required for substrate recognition.** We found no base-specific recognition of RNA by the protein, indicating that the D1D2 core recognizes phosphate backbone and 2′-OH groups only. When the sequences of the two RNA strands are ignored, our D1D2:dsRNA:D1D2 structure appears to be highly symmetric (Fig. 1b). Therefore, we can illustrate protein–RNA interactions with one protein molecule removed (Fig. 2a). As highlighted in the zoom-in boxes, NTE residue S181 and D1 residues S183, T201 and R202 recognize nucleotides 11 and 12 in the central region of RNA Strand 1 via hydrogen bonds to both 2′-OH and phosphate backbone of each nucleotide (Fig. 2b), whereas D2 residues K451, T498, G473, and R480 recognize

### Table 1 Data collection and refinement statistics

| Data collection[a] | |
|---|---|
| Space group | $P3_1$ |
| Cell dimensions | |
| $a, b, c$ (Å) | 66.10, 66.10, 230.46 |
| $\alpha, \beta, \gamma$ (°) | 90, 90, 120 |
| Resolution (Å) | 40.00–2.50 (2.59–2.50)[b] |
| $R_{pim}$ | 0.092 (1.044) |
| $I/\sigma I$ | 8.8 (0.9) |
| Completeness (%) | 96.4 (99.0) |
| Redundancy | 3.8 (4.0) |
| Refinement | |
| Resolution (Å) | 35.91–2.50 (2.64–2.50) |
| No. of reflections | 37,345 (5,464) |
| $R_{work}/R_{free}$ | 0.212/0.246 (0.301/0.353) |
| No. of atoms | |
| Protein | 6,660 |
| Ligand/ion | 1,013 |
| Solvent | 72 |
| $B$-factors | |
| Protein | 58.5 |
| Ligand/ion | 84.4 |
| Solvent | 56.9 |
| R.m.s. deviations | |
| Bond lengths (Å) | 0.004 |
| Bond angles (°) | 0.70 |
| Ramachandran plot | |
| Favored (%) | 96.9 |
| Allowed (%) | 3.0 |
| Disallowed (%) | 0.1 |

[a]A single crystal was used for data acquisition
[b]Values for the highest resolution shell are shown within parentheses

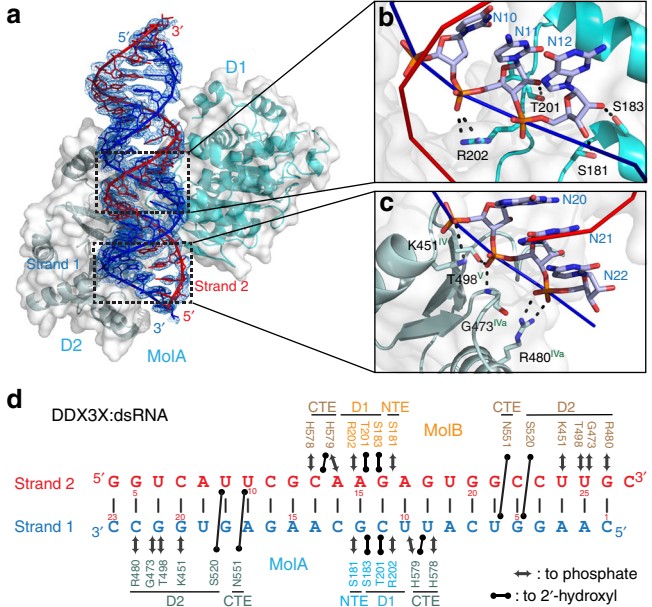

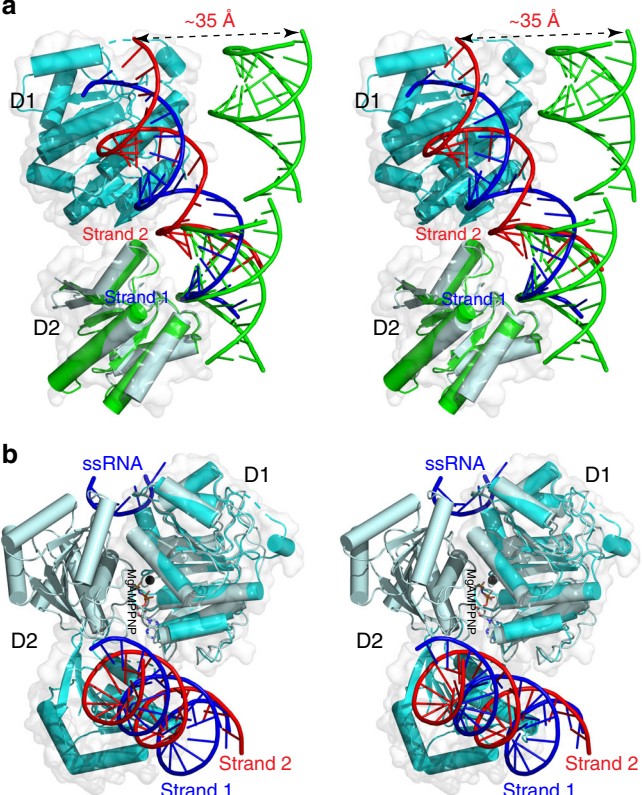

**Fig. 2** Recognition of double-stranded RNA (dsRNA) by the D1D2 core of human DDX3X. **a** Illustration of major protein–RNA interactions with one D1D2 core (MolA, ribbon diagram in cyan/palecyan, overlapped with a transparent molecular surface) and dsRNA (Strands 1 and 2 in blue/red, overlapped with composite omit 2m*Fo*-D*Fc* electron density map contoured at 1σ). MolB is not shown for clarity. **b** D1 recognizes both the 2′-OH groups and phosphate backbone of nucleotide residues 11 and 12 in the central region of RNA Strand 1. **c** D2 mainly recognizes the phosphate backbone of nucleotide residues 20–22 near the 3′ end of the same RNA strand. **d** Schematic representation of protein–RNA interactions observed in the D1D2:dsRNA:D1D2 structure

**Fig. 3** Substrate recognition by the D1D2 core of DEAD-box helicases. **a** In stereo, the superimposition of the D2:dsRNA complex of Mss116p (in green, PDB entry 4DB2) with the D1D2:dsRNA:D1D2 complex of DDX3X (D1 in cyan, D2 in palecyan, RNA strands in red and blue, this work). The two structures are aligned based on D2 and are illustrated as cartoon models. The D1D2 core of DDX3X is also outlined with a transparent molecular surface in white. The double-headed arrow indicates a distance of ~35 Å. **b** In stereo, the superimposition of the D1D2:ssRNA:MgAMPPNP complex of Vasa (protein in palecyan, RNA in blue, Mg in black, and AMPPNP in atomic color; PDB: 2DB3) with the D1D2:dsRNA:D1D2 complex of DDX3X (protein in cyan, RNA Strand 1 in blue and Strand 2 in red; this work). The two structures are aligned based on D1 and are illustrated as cartoon models. The D1D2 core of DDX3X is also outlined with a transparent molecular surface in white. RNA Strand 1 (in blue) is recognized by both D1 and D2 in the two structures. The single-stranded RNA (ssRNA) in the D1D2:ssRNA:MgAMPPNP structure (PDB: 2DB3) corresponds to part of the Strand 1 in the D1D2:dsRNA:D1D2 structure (this work)

nucleotides 20, 21, and 22 near the 3′ end of the same strand (Strand 1) via hydrogen bonds to phosphate backbone only (Fig. 2c). While CTE residue N551 recognizes a 2′-OH group from Strand 2 (Supplementary Fig. 3a), CTE residues H578 and H579 recognize nucleotides 9 and 10 of Strand 1 (Supplementary Fig. 3b). We summarize the protein–RNA interactions in Fig. 2d, showing that D2 also recognizes Strand 2 via residue S520 (Supplementary Fig. 3a). Like other DDXs, DDX3X does not unwind DNA duplex[22]. We show here that all four domains (NTE, D1, D2, and CTE) recognize 2′-OH groups and thereby discriminate against DNA substrates (Fig. 2d). We also show that MolA mainly interacts with Strand 1 and MolB mainly interacts with Strand 2. Together, the two D1D2 cores recognize the 2-turn, A-form dsRNA, which is 22 bp in length (Fig. 2d).

As mentioned earlier, crystal structure of D2 from fly Mss116p in complex with a 14-bp dsRNA was reported previously (PDB: 4DB2)[20]. In the crystallographic asymmetric unit, three 14-bp dsRNA form a pseudo-continuous RNA duplex with four D2s bound on either side of the duplex (the D2:D2:pseudo-dsRNA: D2:D2 ensemble)[20]. Like the DDX3X D2 that recognizes three nucleotides of RNA Strand 1 via hydrogen bonds to phosphate backbone groups (Fig. 2d), the Mss116p D2 also recognizes three nucleotides of Strand 1 via hydrogen bonds to phosphate backbone atoms[20]. Unlike the DDX3X D2 that recognizes a 2′-OH group from Strand 2 through an interaction with the S520 side chain (Fig. 2d), the Mss116p D2 does not interact with Strand 2, although the serine residue is conserved (S455 in Mss116p)[20]. Superposition of the D1D2:dsRNA:D1D2 assembly and the D2:D2:pseudo-dsRNA:D2:D2 ensemble on the basis of one D2 in each structure shows that the RNA duplex in the Mss116p:dsRNA complex is not positioned for D1 binding

(Fig. 3a). Superposition of the D1D2:dsRNA:D1D2 assembly and the D2:D2:pseudo-dsRNA:D2:D2 ensemble on the basis of one D2 in each structure shows dramatically different arrangements of RNA and protein domains (Supplementary Fig. 4). Our structure suggests that if the D1 of Mss116p were present, it would bind Strand 1, and the arrangement of RNA and protein domains could be dramatically improved.

Unlike the Mss116p D2:D2:pseudo-dsRNA:D2:D2 structure (PDB: 4DB2)[20], which was determined in the absence of D1, the structure of Vasa in complex with ssRNA and MgAMPPNP was determined for the D1D2 core (D1D2:ssRNA:MgAMPPNP; PDB: 2DB3)[19]. Human DDX3X and fly Vasa are close orthologs (Supplementary Fig. 1). Superposition of our structure with the Vasa D1D2:ssRNA:MgAMPPNP structure visualizes the unwinding process, that is, the transition from pre-unwound state to post-unwound state. As show in Fig. 3b, the RNA Strand 1 (in blue), which is recognized by both D1 and D2 in the pre-unwound state, is still recognized by both D1 and D2 in the post-unwound state. The

D2 uses the same RNA-binding site in both pre- and post-unwound states and it rotates by ~180° during the transition between the two states. Taken together, the integrity of the D1D2 core is critically important for substrate recognition by DDXs.

**Two DDXs unwind dsRNA cooperatively.** For the D1D2 core of DDX3X (residues 132–607), it was previously shown that RNA duplex unwinding under pre-steady-state conditions results in a sigmoidal functional binding isotherm, indicating a cooperative behavior of the enzyme[14]. We repeated the experiment and observed a similar functional binding isotherm as previously reported (Supplementary Fig. 5a). Furthermore, we calculated the Hill coefficient ($H$) and found that $H = 2.1 \pm 0.3$, indicating a cooperation between two D1D2 cores, in agreement with our structure of pre-unwound D1D2:dsRNA:D1D2 assembly.

Since ATP binding unwinds dsRNA in a cooperative manner, subsequent ATP hydrolysis that facilitates product release should also reflect the cooperative behavior between the two DDX molecules. Driven by this hypothesis, we measured the dsRNA-dependent ATPase activity of the D1D2 core at pre-steady-state, following the protocol previously described for Ded1p[6]. Functional binding isotherm for the ATPase activity was similar to that for the unwinding activity, with the Hill coefficient $H = 1.9 \pm 0.2$ for a 22-bp dsRNA (Fig. 4a) and $1.7 \pm 0.3$ for a 16-bp substrate (Fig. 4b). Together, the results of our Hill cooperativity analysis provide direct evidence for RNA duplex unwinding by two DDXs in a cooperative manner.

**The mechanism of RNA duplex unwinding by DEAD-box helicase.** In general, the functional cycle of DDX can be described with the apo (apo-DDX), pre-unwound (DDX:dsRNA), and post-unwound (DDX:ssRNA:ATP) states. Previously, crystal structure was reported for the D1D2 core of DDX3X at the apo state (PDB: 5E7I)[14]. Here, our D1D2:dsRNA:D1D2 structure at 2.5-Å resolution represents the three-dimensional structure of DDXs at the pre-unwound state. However, a structure of DDX3X at the post-unwound state is still not available. Since DDX3X and Vasa are closely related orthologs (Supplementary Fig. 1), we use the Vasa:ssRNA:AMPPNP structure (PDB: 2DB3)[19] to represent the DDX3X:ssRNA:ATP complex at the post-unwound state. As shown in Fig. 5a, the conformational change of the D1D2 core that occurs during transition between the three states is dramatic. From the apo to pre-unwound state, D2 not only shifts ~60 Å but also rotates by ~180°; from the pre-unwound to post-unwound state, D2 shifts ~35 Å and rotates by ~180°; and from the post-unwound to apo state, D2 shifts ~25 Å without obvious rotation (Fig. 5a).

Our DDX3X:dsRNA structure at the pre-unwound state completes the functional cycle of DDX (Fig. 5b). As described above, the binding of dsRNA to apo-DDX triggers dramatic conformational changes of the enzyme to form the pre-unwound DDX:dsRNA:DDX complex. A total of 26 hydrogen bonds formed between the two D1D2 cores and dsRNA to stabilize the pre-unwound complex (Fig. 2d). In addition, the buried surfaces between the two D1s (~350 Å²), between the D1 pair and dsRNA (~710 Å²), and between each D2 and the corresponding RNA strand (~560 Å²) further stabilize the pre-unwound assembly. ATP binding to the DDX:dsRNA:DDX complex triggers dramatic conformational changes of the protein, which unwinds the dsRNA when each DDX pulls an RNA strand apart from the other, resulting in the post-unwound DDX:ssRNA:ATP complex. At the post-unwound state, ATP hydrolysis triggers a third wave of conformational changes, resulting in the ejection of ssRNA from DDX (Fig. 5b). Structural information is available to describe an additional step in the functional cycle. At least seven more structures were previously determined for truncated DDX3X proteins[14,26–29], among which is the structure of D1D2 in complex with ADP (D1D2:ADP; PDB: 4PXA)[28]. The D1D2:ADP structure and apo-D1D2 (PDB: 5E7I)[14] share the same overall conformation, suggesting free nucleotide release from its binding site. We use the D1D2:ADP structure to represent a post-release state, regarding the release of RNA product in consistence with the pre- and post-unwound nomenclature (Fig. 5b).

**Discussion**

In this study, we report the crystal structure of the D1D2 core of DDX3X in complex with a 2-turn dsRNA. We also observe this D1D2:dsRNA:D1D2 assembly in solution. The observation of a structure in both crystal and solution is biologically significant for obvious reasons. Consistent with the stoichiometry of the structure, the results of our Hill cooperativity analysis of dsRNA-unwinding and ATP-hydrolyzing activities indicate that two DDX molecules function cooperatively. Therefore, our structure at the pre-unwound state of DDX provides new insights into substrate recognition and unwinding by the enzyme. It was previously shown that DDX3X effectively binds and unwinds substrates with 13–19-bp substrates in vitro[22]. Our structure shows that two D1D2 cores jointly reach both terminal ends of a 2-turn, A-form dsRNA (Fig. 1b), explaining why DDXs, which function in a non-progressive manner, rarely unwind RNA duplexes that are longer than two helical turns[30]. Effective unwinding of substrates requires an unpaired 3′ region and a similar cooperative pattern of unwinding was observed for several more substrates[22]. A schematic model suggesting how a 16-bp RNA with a 3′ overhang is recognized by two D1D2 cores is shown (Supplementary Fig. 5b).

We show that the recognition of dsRNA by D2 in the presence or absence of D1 is significantly different (Fig. 3a). In the absence of D1, D2 recognizes Strand 1 only. In the presence of D1, D2 also recognizes Strand 2. This observation indicates that dsRNA recognition and unwinding mechanism is different from that previously proposed based on the D2:dsRNA structure (PDB: 4DB2)[20]. The D2:dsRNA structure suggested that DDX functions as a monomer, that D1 acts as an ATP-binding domain, that D2 acts as an RNA recognition domain, and that conserved motifs in D1 promote dsRNA unwinding by excluding one RNA strand while bending the other[20]. In contrast, our D1D2:dsRNA:D1D2 structure and functional data showed that two DDX molecules function cooperatively, that D1 binds ATP and also recognizes one RNA strand, that one D1D2 core mainly binds RNA Strand 1 and the other D1D2 core mainly binds RNA Strand 2 (Fig. 2d), and that the two dsRNA-bound D1D2 cores undergo dramatic conformational changes upon the binding of MgATP and thereby unwind dsRNA (Fig. 5).

As the closest orthologs, DDX3X and Ded1p represent a DDX subfamily[2]. Hill cooperativity analysis of full-length Ded1p for dsRNA unwinding indicate a three-molecule cooperativity, whereas that for ATP hydrolysis indicate a two-molecule cooperativity[6], suggesting that the third DDX does not hydrolyze ATP. When the minimal functional D1D2 core of DDX3X was redefined, it was indicated that the third DDX molecule increases unwinding efficiency by the fact that full-length DDX3X exhibits much higher unwinding efficiency than the D1D2 core (residues 132–607)[14]. It is known that full-length DDX3X exhibits a three-molecule cooperativity[22], whereas the Hill coefficient for the unwinding activity of the D1D2 core was not reported[14]. We repeated the Hill cooperativity analysis for the D1D2 core under the condition as described[14] and found that the D1D2 core exhibits a two-molecule cooperativity (Supplementary Fig. 5). We also carried out Hill cooperativity analysis for the ATPase activity of the

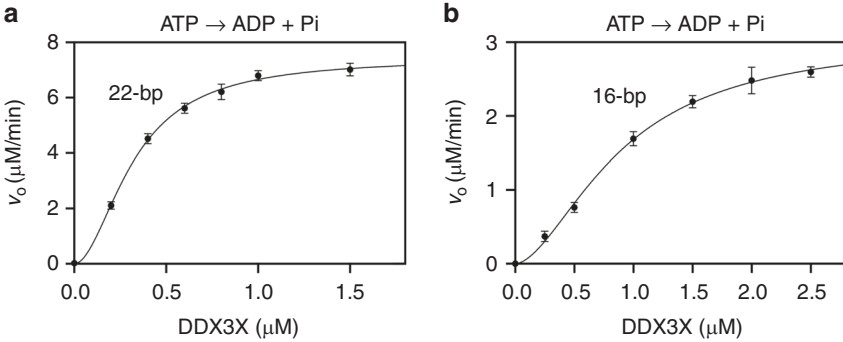

**Fig. 4** Hill cooperativity analysis of the ATPase activity of human DDX3X. **a** ATPase activity of DDX3X was stimulated by 100 nM 22-bp double-stranded RNA (dsRNA) with a 25-nt 3′ overhang (Strand 1: 5′-UCGUGGCAUUUCUGCGUCGUUCUUUUCUUUUCUUUUCUUUUCUUUUC-3′; Strand 2: 5′-GAACGACGCAG AAAUGCCACGA-3′) at indicated concentrations of the helicase. Solid line is the best fits to the Hill equation ($V_{max} = 7.4 \pm 0.3$ μM min$^{-1}$, $K_{1/2} = 320 \pm 20$ nM, $H = 1.9 \pm 0.2$). **b** ATPase activity of DDX3X was stimulated by 100 nM 16-bp dsRNA with a 25-nt 3′ overhang (Strand 1: 5′-GCGUCUUUACGGUGCUU AAAACAAAACAAAACAAAACAAAA-3′; Strand 2: 5′-AGCACCGUAAAGACGC-3′) at indicated concentrations of the helicase. Solid line is the best fits to the Hill equation ($V_{max} = 3.1 \pm 0.4$ μM min$^{-1}$, $K_{1/2} = 895 \pm 120$ nM, $H = 1.7 \pm 0.3$). Error bars represent 1 SD. Source data are provided as a Source Data file

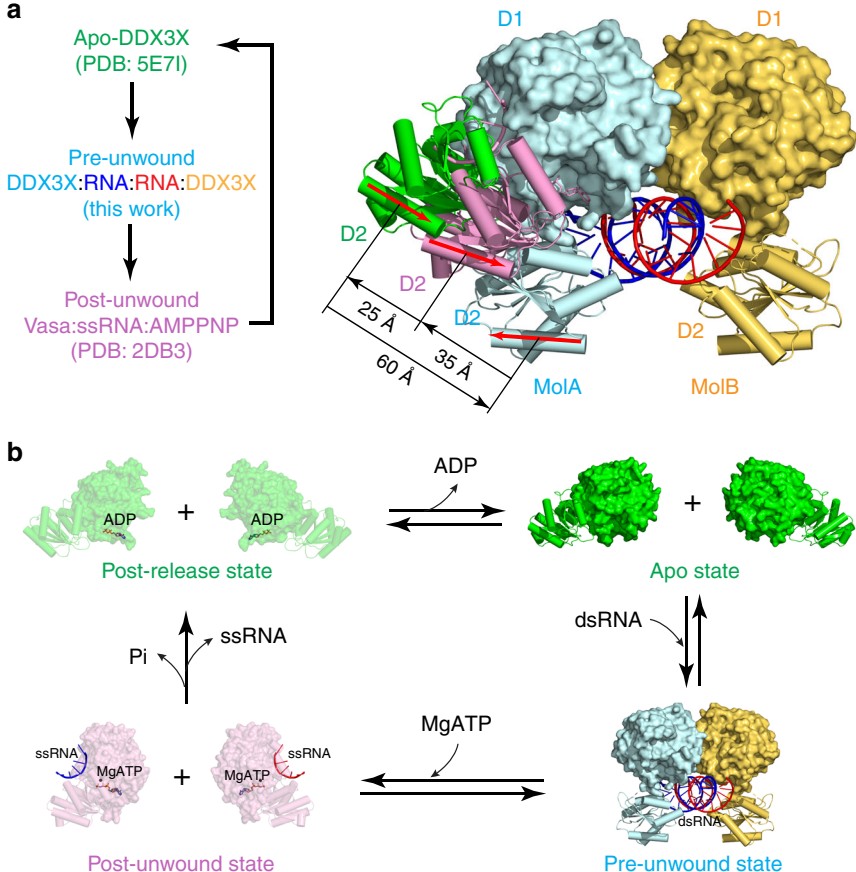

**Fig. 5** Unwinding of double-stranded RNA (dsRNA) by DEAD-box helicases. **a** Conformational difference between the apo-DDX3X (in green, PDB: 5E7I), DDX3X:dsRNA:DDX3X (in cyan and orange, this work), and Vasa:MgAMPPNP:ssRNA (in pink, PDB: 2DB3) structures. The structures are aligned based on Cα positions in D1. The D1 of the DDX3X:dsRNA:DDX3X structure is illustrated as a molecular surface, and those of other structures are not shown for clarity. The D2 of all three structures are shown as cartoon models (helices as cylinders, strands as arrows, and loops as tubes). The orientation of D2 relative to the fixed D1 at each state are indicated with a red arrow on the same α-helix in D2. **b** Represented by the apo (PDB: 5E7I), pre-unwound (this work), post-unwound (PDB: 2DB3), and post-release (PDB: 4PXA) states, the functional cycle explains a four-step mechanism of RNA duplex unwinding by DEAD-box helicases

D1D2 core and found that the D1D2 core also exhibits a two-molecule cooperativity (Fig. 4). Together, our results of Hill cooperativity analysis, which are consistent for both dsRNA-unwinding and ATP-hydrolyzing activities, together with our

D1D2:dsRNA:D1D2 structure, which is observed in both crystal and solution, demonstrate that two DDX molecules are required to unwind an RNA duplex with the consumption of two ATP molecules. Participation of a third molecule optimizes

the unwinding efficiency, which is only observed for the full-length DDX3X and Ded1p[6,14,22].

Members of the DDX3X/Ded1p subfamily feature a low complexity region in their C-terminal sequences of ∼60 amino acid residues[2]. Removing 69 residues from the C terminus of Ded1p, the truncated DDX forms dimers instead of forming the trimer, suggesting that the low complexity region is part of the protein–protein interface in the trimer[6]. With 55 C-terminal residues of DDX3X removed, the truncated DDX exhibits two-molecule cooperativity instead of exhibiting three-molecule cooperativity. The three-molecule cooperativity described for Ded1p features a division of labor (RNA-loading and RNA-unwinding) between the three protomers[6]. We believe that this is also true for the three-molecule cooperativity of DDX3X, suggesting that the third protomer plays the RNA-loading role and thereby optimizes the RNA-unwinding efficiency. It is clear that the low complexity region is required for the recruitment of the third protomer, but how the third protomer associates with the other two via the low complexity region remains to be elucidated.

Members of the same enzyme family often adopt the same basic mechanism of action. Our unwinding mechanism of DDX3X provides a model for mechanistic studies of other DDX proteins, especially members of the DDX3/Ded1 and DDX4/Vasa subfamilies. This will become clear when high-resolution structures of the D1D2 cores of other DDXs at the pre-unwound state become available.

## Methods

**Expression and purification of human DDX3X.** The minimal functional D1D2 core of DDX3X (residues 132–607, Fig. 1a) was cloned into the pDEST-527 vector with a His$_6$-MBP tag at the N terminus, using primers 5′-GGGGACAACT TTGTACAAAAAAGTTGTGGAGAACCTGTACTTCCAGGGTCCAGT TGAGGCAACAGGCAA-3′ and 5′-GGGGACAACTTTGTACAAGAAAGT TGCATTAACCGCTACTTTGTCGGTAGT-3′. The clone was over-expressed in *Escherichia coli* strain BL21-CodonPlus(DE3)-RIL (Agilent). Culture was grown to mid-log phase at 37 °C in Luria-Bertani medium, induced by the addition of β-D-1-thiogalactopyranoside to a final concentration of 1 mM. The cells were incubated for additional 4 h, harvested by centrifugation, and stored at −80 °C until purification.

Protein purification was performed at 4 °C using an AKTA chromatography system with prepacked columns (GE Healthcare). Cells were suspended in buffer A [30 mM Tris-HCl (pH 7.5), 1 M NaCl, 10% (v/v) glycerol, 1 mM dithiothreitol (DTT), and 25 mM imidazole] supplemented with complete EDTA-free protease inhibitor cocktail tablets (Roche Molecular Biochemicals), lysed using an APV-1000 homogenizer (Invensys APV Products), and centrifuged at 26,892 × g for 30 min. The supernatant was applied onto a 5-ml His Trap FF column pre-equilibrated in buffer A. The column was washed with buffer A to baseline and the fusion protein was eluted with buffer B [30 mM Tris-HCl (pH 7.5), 1 M NaCl, 10% (v/v) glycerol, 1 mM DTT, and 400 mM imidazole]. The fusion protein was digested overnight with 0.2 mg ml$^{-1}$ TEV protease. The digest was buffer-exchanged to buffer A and applied onto a 10-ml His Trap FF column pre-equilibrated with buffer A. The target protein was isolated in the column flow-through and was concentrated to a desired volume (<10 ml), and then applied onto a Superdex 200 gel filtration column pre-equilibrated in buffer C [25 mM Tris-HCl (pH 7.5), 200 mM NaCl, and 1 mM DTT]. The protein was collected from peak fractions and its quality was analyzed by sodium dodecyl sulfate gel electrophoresis.

**Crystal structure determination.** The RNA oligo was purchased (GE Dhamacon Inc.), annealed without further purification, and used in crystallization. The purified DDX3X protein (6 mg ml$^{-1}$) was mixed with dsRNA in a molar ratio of 2:1 and incubated on ice for 30 min. Crystals were grown by mixing the complex solution with an equal volume of reservoir solution [0.2 M magnesium chloride, 0.1 M Tris (pH 8.5), and 25% (v/v) PEG 3350], using the sitting-drop vapor-diffusion method. For data collection, the crystals were soaked in the reservoir solution containing 20% (v/v) ethylene glycol and flash frozen in liquid nitrogen. X-ray (λ = 1.0 Å) diffraction data were collected at −173 °C using a Rayonix MX300HS detector at the SER-CAT 22-ID beamline of the Advanced Photon Source (APS) and processed using the HKL-3000 program suite[31]. Initially, the data were scaled in space group P3$_1$21. The structure was solved by molecular replacement using the PHASER program[32] with the apo-DDX3X structure (PDB: 5E7I) as the search model. Programs Coot[33] and Phenix[34] were used for model building and structure refinement. The resulting structure contains one peptide chain and one RNA strand in the asymmetric unit. During model building and refinement, we discovered the asymmetric binding of the dsRNA by the protein

(Supplementary Fig. 2a), rescaled the data in P3$_1$, and finished the refinement. The final structure was evaluated by the validation server of wwPDB[35]. Data collection and structure statistics are summarized in Table 1.

**SAXS analysis.** The SAXS data were recorded at the APS beamline 12-ID-B, using a PILATUS 1M detector (Dectris) for SAXS and a PILATUS 100k detector (Dectris) for wide-angle X-ray scattering. The wavelength λ of X-ray radiation was 0.8856 Å and the momentum transfer q was recorded in a range of 0.006–2.8 Å$^{-1}$ [$q = (4\pi/\lambda)\sin\theta$, where 2θ is the scattering angle]. The DDX3X:dsRNA complex (4 mg ml$^{-1}$) was pre-equilibrated in a SAXS buffer [50 mM HEPES (pH 7.5), 100 mM NaCl, and 2 mM β-mercaptoethanol] for data collection. The scattering profiles were measured in three solute concentrations (4-, 2-fold dilution and stock solution) to remove the scattering contribution due to inter-particle interactions and to extrapolate the data to infinite dilution. A set of 30 two-dimensional images were recorded for each buffer or sample solution using a flow cell, with the exposure time of 1–2 s per image to minimize radiation damage and optimize signal-to-noise ratio. The images were reduced on site to one-dimensional scattering profiles using the Matlab software package (The MathWorks Inc., Natick, MA, USA).

The SAXS data was analyzed using IGOR-Pro (WaveMetrics) and the ATSAS suite[36] in a similar way as described[37]. Briefly, the forward scattering intensity I(0) and the radius of gyration (R$_g$) were calculated from the infinite dilution data at low q values in the range of qR$_g$<1.3, using the Guinier approximation in PRIMUS[38]. These parameters were also estimated from the scattering profile with a broader q range of 0.006–0.30 Å$^{-1}$ using the indirect Fourier transform method implemented in program GNOM[39], along with the PDDF and the maximum dimension of the protein (D$_{max}$). The molecular weight was calculated by the SAXS MoW method[40], which is independent of protein concentration. Low-resolution ab initio shape envelopes were calculated using DAMMIN[41], which generates models represented by an ensemble of densely packed beads, using scattering profiles within the q range of 0.006–0.30 Å$^{-1}$. A total of 20 independent models were created and averaged by DAMAVER[25], superimposed by SUPCOMB[42] based on the normalized spatial discrepancy criteria, and filtered using DAMFILT to generate the final model. The theoretical scattering intensity of the crystal structure was calculated and fitted to the experimental scattering intensity using CRYSOL[43]. Data collection and scattering-derived parameters are summarized in Supplementary Table 1.

**RNA duplex unwinding and ATPase assays.** The RNA substrates (Fig. 4, Supplementary Fig. 5) were purchased (Integrated DNA Technology). The RNA-unwinding assay was performed under the same condition as described[14,22]. Briefly, reaction mixtures (30 μl) containing 40 mM Tris-HCl (pH 8.0), 50 mM NaCl, 0.5 mM MgCl$_2$, 2 mM DTT, 1 U ml$^{-1}$ RNasin (Thermo Fisher Scientific), and 0.01% (v/v) IGEPAL. Concentrations of RNA and enzyme are indicated (Fig. 4, Supplementary Fig. 5). Reactions were initiated by adding MgATP (2 mM final concentration). The strand separation was detected using Chemiluminescent Nucleic Acid Detection Module Kit (Thermo Fisher Scientific) and visualized by ChemiDoc Imaging System (Bio-Rad). ATPase measurements were performed under the same condition in a 300 μl reaction mixture. The release of phosphate was measured continuously using EnzChek Phosphate Assay Kit (Thermo Fisher Scientific). Unwinding rates and functional binding parameters were fitted using the PRISM software (GraphPad) as described[22].

**Reporting summary.** Further information on research design is available in the Nature Research Reporting Summary linked to this article.

## Data availability
Structure factors and atomic coordinates have been deposited in the Protein Data Bank (PDB) with the accession code 6O5F. The source data underlying Figs. 1d, 4a, b and Supplementary Fig 5a are provided as a Source Data file. Other data are available from the corresponding author upon reasonable request.

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

## Acknowledgements

We thank Lixin Fan and Xiaobing Zuo for help and advice during the SAXS experiment, and Xianyang Fang, Lan Jin, and Alexander Wlodawer for reading the manuscript and discussions. We acknowledge the use of SAXS core facility of National Cancer Institute, Center for Cancer Research. SAXS data were collected at APS beamline 12-ID-B, Argonne National Laboratory (ANL). X-ray diffraction data were collected at the SER-CAT beamlines of APS, ANL. Use of the APS was supported by the US Department of Energy under Contract No. DE-AC02-06CH11357 and under the Partner User Proposal PUP No. 22978. This research was supported by the Intramural Research Program of the NIH, National Cancer Institute, Center for Cancer Research.

## Author contributions

H.S. and X.J. designed the experiments. H.S. performed the experiments. H.S. and X.J. analyzed the data and wrote the manuscript.

## Additional information

**Competing interests:** The authors declare no competing interests.

