## [Peer Review File · Nature Communications]

Reviewers' comments:

Reviewer #1 (Remarks to the Author):

The manuscript by Song and Ji describes the first crystal structure of the DDX3X:dsRNA complex in a pre-unwound state. Based on this newly determined structure the authors claim to provide the missing "structural element" required to explain the mechanism of dsRNA unwinding by DDX3X helicase.

However, the postulated mechanism requires additional confirmation in order to be considered as plausible. First of all, in contrast to the previous work, the authors unexpectedly suggest that two DDX3X molecules are needed to unwind the dsRNA. Consequently, the previous biochemical experiments suggesting one ATP binding event for unwinding seem to be not valid any longer. The authors should provide a biochemical evidence that for the co-crystallized dsRNA two ATP binding events are necessary and, in addition, investigate if it could be related to the dsRNA used.

The presented crystal structure of two DDX3X helicases bound to dsRNA molecule could also be interpreted as DDX3X:ssRNA - ssRNA-DDX3X complex, where "-" symbolizes a symmetry axis. In other words, in a higher symmetry space group the asymmetric unit would be occupied by one DDX3X molecule bound to a ssRNA (which could be a part of a dsRNA of course). This scenario should be considered as a perfect twinning (h,-h-k,l twin operator) has been detected during the validation performed by the PDB. The estimated twinning could in principle be a result of a merohedral twinning or too low symmetry used. The latter case seems very plausible due to the author's statement: "Our DDX3X:dsRNA structure is highly symmetric (Fig. 1b). Therefore, we can illustrate protein-RNA interactions with one protein molecule removed (Fig. 2a)". However, without additional analysis it is not possible to judge, and unfortunately, the authors do not provide any details concerning twinning and refinement strategy (in case of twinning).

Summarizing, this structure of the pre-unwound state of a DEAD-box helicase is indeed an important structural snapshot needed to complete the repertory of functional states in the DEAD-box helicase family. However, the gain of mechanistic insights by this structure is limited and it even raises more questions than answers, which are not appropriately addressed by the authors. The claimed cooperativity of the D1 domains cannot be satisfyingly proven by this structure and additional experiments are needed to validate this assumption. The proposed disagreements with current models also need more in-depth analysis. Due to the limited gain of mechanistic insights and the rather weak evidence for the assumptions made in this manuscript, additional functional data should be provided to justify a publication in Nature Communications.

Specific comments:

- Page 2 In the Abstract, the authors claim to find that D1 does not only interacts with RNA, but also discriminates DNA from RNA. This finding has been already described in the literature, see e.g. [8] for details, and could be removed from the Abstract.

- Page 2 "...DDX3X is among the most comprehensively studied DDXs that yielded structures of the DDX core, either RNA free 8-13 or in complex with a single RNA strand at post-unwound state 14-18."

→ Sentence is not completely true. There is no structure of DDX3X in complex with ssRNA in the PDB and none of the citations (14-18) is about DDX3X. There are ssRNA complex structures only for other DEAD-box members.

- Page 2 The word "cooperative" is constantly used throughout the manuscript. The structure alone is not sufficient to describe a cooperative effect. Cooperativity implies an effect (be it positive or negative) of one DDX3X molecule on the other, but the structure merely shows that the D1 domains share a contact surface. The structure does not give a definite proof if really both molecules are needed to unwind the double strand or if the work could be done as well by one of the DDX3X molecules.

To describe such a cooperativity additional functional assays are needed. Structure-based mutants impairing the D1 interaction could be tested for helicase activity. Such an experiment would also

rule out that the D1 interaction surface displays a crystal artifact.

- Page 3 „In contrast, no structural information is available at the pre-unwound state for any DDX core; hence, it is not clear how DDX recognizes and unwinds dsRNA.“

→ Again, this is not completely true. There is a partial structure of the pre-unwound state with the D2 domain bound to dsRNA (PDBid: 4db2). It would also have been worth mentioning, that the interaction pattern of this domain is almost identical in both structures. Thus, a valid partial model for recognition was already described previously.

-Page 3 „ Intriguingly, our structure shows that the dsRNA was recognized and bound by two DDX3X cores.“

→ The fact that two molecules could be able to bind simultaneously on a dsRNA helix is not completely surprising. The Mss116p-D2-dsRNA complex structure (PDBid: 4db2) shows two D2 domains bound and superimposing a partially open structure (PDBid: 2i4i) one can clearly see, that the D1 domain could fit between the two D2 domains in a comparable manner as seen in the presented structure. However, the orientation of the second D2 domain differs in both structures. This should be highlighted and the manuscript would greatly improve if the similarities but also the differences would be discussed.

Page 5 “Although it may not as stable as the partially closed structure, the partially open structure co-exists with the former in an equilibrium.”

→ Is there any proof for this assumption? If no experiments or citations can be provided, please make clear that it is an assumption to help to describe your findings and not a given fact.

Page 6 “Our DDX3X:dsRNA structure indicates that two DDXs are needed to reach both terminal ends of a 2-turn, A-form dsRNA (Fig. 1b), indicating that unwinding such a dsRNA requires two ATP binding events. This observation agrees well with a previous observation that one ATP binding event can open a dsRNA of 9-12 bp in length.”

→ Sharma et al. (2017) show that DDX3X is able to unwind duplexes as short as 13bp (almost half of the length of the dsRNA used in this structure). As the contacting D1 domains span roughly the entire length of the 23bp dsRNA and assuming this “cooperative” interaction is required as described in this manuscript, it can not explain how a short 13bp dsRNA can be disrupted. Two DDX3X molecules could not fit on the 13bp dsRNA in a way that the D1 domains could still contact each other as seen here.

Page 19 → PDB entry 2dbd is wrong, should be 2db3

Figure 2. The presented electron density map is described as 2Fo-Fc. This electron density map could be strongly biased towards model, therefore the authors should present a less biased map, e.g.: composite omit 2mFo-DFc map, which can be easily calculated with Phenix.

Reviewer #2 (Remarks to the Author):

This paper presents the crystal structure of the catalytic core of the DEAD-box helicase DDX3X bound to dsRNA as well as small angle scattering data for the complex. The structure present the first structural insight into duplex RNA binding by this enzyme. One feature that the structure illuminates is the basis for discrimination between DNA and RNA by DDX3X (D1 and CTE bind multiple 2'OH on the backbone) and the cooperativity of molecules during unwinding (two symmetrical cores of DDX3X bind the RNA together). Alongside the structure, the authors propose a complete unwinding cycle which includes three previously solved structures of the DDX3X core. The structure provides insight into the catalytic cycle of the RNA helicase DDX3X and substrate recognition. The structures are of high quality and the analysis is thorough. There are a few small points that hopefully will be helpful for the authors.

On page 4 paragraph 2, it is noted that the CTW also interacts with the RNA strand's 2'-OH groups, however, the interaction itself is not shown. Instead, a cartoon schematic indicates that the interaction exists. It would be helpful to see the interaction and not just the schematic, if possible.

On page 5, at the end of paragraph 1, there exists a discrepancy between the figure and the text. The text states that "... the closed (post-unwound) and partially closed (apo) states are similar, while the open (pre-unwound) and partially open (apo) states are similar." In figure 3a, the partially open figure is attributed to the AMP bound DDX3X, and not the apo state.

Please elaborate on the partially open state, whether or not AMP binding is used to mimic the apo state and whether AMP binding changes the shape of the molecule from closed to open state.

Figure 3 is confusing. It's unclear why these states are necessary as part of the unwinding cycle. Also, it's not clear from the figure where the RNA is located in the closed state.

Page 12 "Data availability. Structure factors and atomic coordinates will be deposited in the Protein Data Bank (PDB) and released upon publication of this manuscript." This is not good practice – a pdb validation report really must be provided as a condition for publication and would have helped with the review.

Minor corrections

Page 4 "Our structure shows that D1 and CTE discriminate DNA from RNA, which was never thought of before." This should be rephrased – maybe instead write 'not previously recognised or demonstrated'.

Page 5, 2nd line 2nd paragraph "Although it may not be as ..."

Page 8, reference 8 incomplete

Page 11, methods, half way down the page, need centrifugation in force, not rpms.

Figure 3b – typo in label for figure "Partially Open state"

We would like to thank the reviewers for their insightful comments and helpful advice.

Please see our point-to-point response (in blue) to the referees' comments below.

Reviewers' comments

Reviewer #1 (Remarks to the Author):

The manuscript by Song and Ji describes the first crystal structure of the DDX3X:dsRNA complex in a pre-unwound state. Based on this newly determined structure the authors claim to provide the missing “structural element” required to explain the mechanism of dsRNA unwinding by DDX3X helicase.

However, the postulated mechanism requires additional confirmation in order to be considered as plausible. First of all, in contrast to the previous work, the authors unexpectedly suggest that two DDX3X molecules are needed to unwind the dsRNA. Consequently, the previous biochemical experiments suggesting one ATP binding event for unwinding seem to be not valid any longer. The authors should provide a biochemical evidence that for the co-crystallized dsRNA two ATP binding events are necessary and, in addition, investigate if it could be related to the dsRNA used.

Our manuscript was originally submitted to *Nature* in the format of Letters to Nature. Due to the limited size of that format, we were not able to provide enough background information. It was previously shown that RNA unwinding by DDX3X or Ded1p under pre-steady-state conditions resulted in a sigmoidal functional binding isotherm, indicating that DDX3X and Ded1p unwind RNA duplexes, containing 13-19 base pairs (bp), in a cooperative manner¹⁻³. Our DDX3X D1D2:dsRNA:D1D2 structure, not only seen in crystal but also observed in solution, indicates that two DDXs cooperatively unwind dsRNA, which is consistent with the functional data.

Previously, Chen and co-workers reported that RNA duplex unwinding needs one ATP binding event⁴. The study was performed by comparing ATP turnover rate and unwinding rate of DNA-RNA hybrids, showing that unwinding a 6-bp substrate needs the hydrolysis of ~0.5 ATP and unwinding an 11-bp dsRNA needs ~1 ATP. Although the trend indicates that more ATP binding and hydrolysis is required for unwinding longer substrates, it is impossible to study RNA duplex longer than one turn using the method as described³. It was shown that, for a 10-bp duplex, 95% ATP binding and hydrolysis event produced strand separation; for a 16-bp duplex, ~30% of binding event was productive; and for a 19-bp substrate, less than 1% result in detectable strand separation³. Using a different approach, Putnam and co-workers determined the number of ATP binding events for RNA duplex unwinding by Ded1p, where it was shown that two ATP binding events are required for strand separation³. The experiment was performed by determining the dsRNA-dependent ATPase activity under pre-steady-state conditions, resulting in a sigmoidal functional binding isotherm with Hill coefficient of ~2. This result is consistent with the result of single-molecule FRET analysis³.

The cooperative behavior of Ded1p was elucidated not only for ATP-hydrolyzing activity, but also for RNA-unwinding activity³. The results for the latter indicated a 3-molecule cooperation event, suggesting that ATPase activity requires cooperation between fewer Ded1p

protomers than unwinding activity³. During the revision, we designed and performed additional Hill kinetics experiments under the pre-steady-state conditions.

Using the minimal functional D1D2 catalytic core of DDX3X, we performed similar dsRNA-dependent ATPase experiments as described². We determined the ATPase activity of D1D2 for 16-bp and 22-bp substrates, resulting in a similar sigmoidal functional binding isotherm and Hill coefficients of ~2 (Fig. 4). Our DDX3X data is consistent with the Ded1p data³, showing that dsRNA-dependent ATPase activity is cooperative between two DDX molecules. Using the same D1D2 core of DDX3X, we also performed the RNA unwinding experiment. For the same 16-bp substrate, we observed a similar cooperative isotherm with a Hill coefficient of ~2 (Extended Data Fig. 5). As such, our Hill kinetics results for both ATPase assay and unwinding assay are consistent, which further validated our D1D2:dsRNA:D1D2 structure. Along with Fig. 4 and Extended Data Fig. 5, a new section, entitled “**Hill kinetics of DDX3X activities indicate that two DDXs function cooperatively**,” is added in **Results** (Page 8, Lines 1-16).

Together, our new and previous data provide a comprehensive explanation for the cooperative nature of DDX action, which is explained in Discussion (Page 11, Line 8 – Page 12, Line 2). The dsRNAs used in these functional and structural analyses are 16-26 base pairs long, with a 3' overhang on either one or two terminal ends. The results are consistent and sequence specificity has not been observed.

The presented crystal structure of two DDX3X helicases bound to dsRNA molecule could also be interpreted as DDX3X:ssRNA - ssRNA-DDX3X complex, where “-” symbolizes a symmetry axis. In other words, in a higher symmetry space group the asymmetric unit would be occupied by one DDX3X molecule bound to a ssRNA (which could be a part of a dsRNA of course). This scenario should be considered as a perfect twinning (h,-h-k,l twin operator) has been detected during the validation performed by the PDB. The estimated twinning could in principle be a result of a merohedral twinning or too low symmetry used. The latter case seems very plausible due to the author's statement: “Our DDX3X:dsRNA structure is highly symmetric (Fig. 1b). Therefore, we can illustrate protein-RNA interactions with one protein molecule removed (Fig. 2a).” However, without additional analysis it is not possible to judge, and unfortunately, the authors do not provide any details concerning twinning and refinement strategy (in case of twinning).

We thank the reviewer for pointing this out. Initially, the data was scaled in $P3_121$ and the corresponding solution contains one D1D2 core and one RNA strand in the asymmetric unit. During refinement, we discovered the asymmetric binding of dsRNA by the protein (Extended Data Fig. 2a), resulting in different sequences in the two RNA strands bound to the D1D2 cores. We rescaled the data in $P3_1$ and finished the refinement. In the revised version, we have included this in both Results (Page 4, Lines 13-18) and Methods (Page 13, Line 19 – Page 14, Line 3).

During the revision, we also tested refinement with the suggested twin law. The statistics (without the twin law: $R_{work}=0.213$, $R_{free}=0.246$; with the twin law: $R_{work}=0.214$, $R_{free}=0.243$) indicated that the crystal is not twinned.

Summarizing, this structure of the pre-unwound state of a DEAD-box helicase is indeed an important structural snapshot needed to complete the repertory of functional states in the DEAD-box helicase family. However, the gain of mechanistic insights by this structure is limited and it even raises more questions than answers, which are not appropriately addressed by the authors. The claimed cooperativity of the D1 domains cannot be satisfyingly proven by this structure and additional experiments are needed to validate this assumption. The proposed disagreements with current models also need more in-depth analysis. Due to the limited gain of mechanistic insights and the rather weak evidence for the assumptions made in this manuscript,

additional functional data should be provided to justify a publication in Nature Communications.

We thank the reviewer for emphasizing the significance of our new structure. During the revision, we provided additional background information, design and performed Hill kinetics experiments, correlated our new and previous Hill kinetics data, analyzed our new structure together with existing structures, and correlated structural information with Hill kinetics data. During this process, we provided additional in-depth analysis of the proposed disagreements with previous model in Discussion (Page 10, Line 19 – Page 11, Line 7) and explained the discrepancy between the Hill coefficient for the unwinding activity (~ 3) and that for the ATPase activity (~ 2) of full-length DDX (Page 11, Line 8 – Page 12, Line 2).

Specific comments:

- Page 2 In the Abstract, the authors claim to find that D1 does not only interacts with RNA, but also discriminates DNA from RNA. This finding has been already described in the literature, see e.g. [8] for details, and could be removed from the Abstract.

We removed this statement from the abstract.

- Page 2 „...DDX3X is among the most comprehensively studied DDXs that yielded structures of the DDX core, either RNA free 8-13 or in complex with a single RNA strand at post-unwound state 14-18.“

→ Sentence is not completely true. There is no structure of DDX3X in complex with ssRNA in the PDB and none of the citations (14-18) is about DDX3X. There are ssRNA complex structures only for other DEAD-box members.

We thank the reviewer and have made it clear during the revision (Page 2, Line 22 – Page 3, Line 5).

- Page 2 The word “cooperative” is constantly used throughout the manuscript. The structure alone is not sufficient to describe a cooperative effect. Cooperativity implies an effect (be it positive or negative) of one DDX3X molecule on the other, but the structure merely shows that the D1 domains share a contact surface. The structure does not give a definite proof if really both molecules are needed to unwind the double strand or if the work could be done as well by one of the DDX3X molecules.

To describe such a cooperativity additional functional assays are needed. Structure-based mutants impairing the D1 interaction could be tested for helicase activity. Such an experiment would also rule out that the D1 interaction surface displays a crystal artifact.

Following the reviewer’s advice, we designed and performed Hill kinetics experiments, showing that the two D1D2 cores cooperatively unwind RNA duplexes of 16 and 22 bp in their lengths with a 25-nt 3' overhang (Fig. 4). The D1D2 core exists as monomer in both apo and post-unwound states (Fig. 5b). As such, the formation of the pre-unwound complex is mainly promoted by D1D2-dsRNA recognitions the relatively weak interaction between the two D1s. This is consistent with the small buried surface of $\sim 350 \text{ \AA}^2$ between D1s, which is much smaller than the buried surface between the D1D2 core and dsRNA (1830 \AA^2). The D1D2:dsRNA:D1D2 assembly was also observed in solution (by SAXS), demonstrating that the formation of the assembly is not a crystallization artifact.

- Page 3 „In contrast, no structural information is available at the pre-unwound state for any DDX core; hence, it is not clear how DDX recognizes and unwinds dsRNA.“

→ Again, this is not completely true. There is a partial structure of the pre-unwound state with

the D2 domain bound to dsRNA (PDBid: 4db2). It would also have been worth mentioning, that the interaction pattern of this domain is almost identical in both structures. Thus, a valid partial model for recognition was already described previously.

We thank the reviewer for pointing this out. In the revised manuscript, we introduced the Mss116p D2:dsRNA structure in the **Introduction** (Page 3, Lines 3-4), and analyzed the similarity and differences between the D2:dsRNA structure with our D1D2:dsRNA:D1D2 (Fig. 3a, Extended Data Fig. 4; Page 6, Line 19 – Page 7, Line 11).

-Page 3 „ Intriguingly, our structure shows that the dsRNA was recognized and bound by two DDX3X cores.“

→ The fact that two molecules could be able to bind simultaneously on a dsRNA helix is not completely surprising. The Mss116p-D2-dsRNA complex structure (PDBid: 4db2) shows two D2 domains bound and superimposing a partially open structure (PDBid: 2i4i) one can clearly see, that the D1 domain could fit between the two D2 domains in a comparable manner as seen in the presented structure. However, the orientation of the second D2 domain differs in both structures. This should be highlighted and the manuscript would greatly improve if the similarities but also the differences would be discussed.

Again, we appreciate the reviewer's advice very much. Indeed, the manuscript is significantly improved via analyzing the similarities and differences between the D2:dsRNA structure (PDB: 4DB2) and our D1D2:dsRNA:D1D2 structure. Our crystal structure of the D1D2:dsRNA:D1D2 assembly, which is observed in solution and validated by Hill kinetics data, provides the opportunity of in-depth analysis the new structure in light of existing structures. For the two dsRNA-bound structures, we compared not only the interaction between D2 and dsRNA (Fig. 3a), but also the arrangement of RecA-like domains and dsRNA (Extended Data Fig. 4; Page 6, Line 19 – Page 7, Line 11).

Page 5 “Although it may not as stable as the partially closed structure, the partially open structure co-exists with the former in an equilibrium.”

→ Is there any proof for this assumption? If no experiments or citations can be provided, please make clear that it is an assumption to help to describe your findings and not a given fact.

We are grateful to the reviewer for pointing this out. During the revision, we realized that in addition to the assumption of the equilibrium between two apo conformations, representing an apo conformation by the D1D2:AMP structure (PDB: 2I4I)⁵ is also an assumption. Although AMP is not relevant to the unwinding reaction, the D1D2:AMP complex is not ligand free. Therefore, we removed the D1D2:AMP structure and so the comparison of apo, pre-unwound, and post-unwound states became much easier to visualize (Fig. 5a). Also, we included the D1D2:ADP structure (PDB: 4PXA)⁶ In the functional cycle to represent a post-release state, regarding the release of RNA product in consistence with the pre- and post-unwound nomenclature (Page 9, Lines 19 – Page 10, Line 2). The functional cycle is still composed of four states, but without any assumptions (Fig. 5b).

Page 6 “Our DDX3X:dsRNA structure indicates that two DDXs are needed to reach both terminal ends of a 2-turn, A-form dsRNA (Fig. 1b), indicating that unwinding such a dsRNA requires two ATP binding events. This observation agrees well with a previous observation that one ATP binding event can open a dsRNA of 9-12 bp in length.”

→ Sharma et al. (2017) show that DDX3X is able to unwind duplexes as short as 13bp (almost half of the length of the dsRNA used in this structure). As the contacting D1 domains span roughly the entire length of the 23bp dsRNA and assuming this “cooperative” interaction is required as described in this manuscript, it can not explain how a short 13bp dsRNA can be

disrupted. Two DDX3X molecules could not fit on the 13bp dsRNA in a way that the D1 domains could still contact each other as seen here.

As the reviewer pointed out, DDX3X effectively binds to and unwinds substrates with 13-19-bp substrates in vitro. Effective unwinding of those substrates, however, requires a 3' unpaired overhang at one terminal end of the duplex¹. Given that D2 recognizes the termini of dsRNA and do not discriminate ssRNA from dsRNA, a model can be readily derived on the basis of our D1D2:dsRNA:D1D2 structure (Fig. 2d) to illustrate how two D1D2 cores recognize substrates of 13-19-bp in length. A model complex for a 16-bp dsRNA is shown in Extended Data Fig. 5b, from which a model for a 13-bp dsRNA can be derived by removing three base pairs from the terminal end that has the 3' single-stranded overhang.

Page 19 → PDB entry 2dbd is wrong, should be 2db3

We have corrected the typo.

Figure 2. The presented electron density map is described as 2Fo-Fc. This electron density map could be strongly biased towards model, therefore the authors should present a less biased map, e.g.: composite omit 2mFo-DFc map, which can be easily calculated with Phenix.

We thank the reviewer for the advice. We calculated the 2mFo-DFc map, using the composite_omit_map program embedded in Phenix, and presented it in Fig. 2a.

Reviewer #2 (Remarks to the Author):

This paper presents the crystal structure of the catalytic core of the DEAD-box helicase DDX3X bound to dsRNA as well as small angle scattering data for the complex. The structure present the first structural insight into duplex RNA binding by this enzyme. One feature that the structure illuminates is the basis for discrimination between DNA and RNA by DDX3X (D1 and CTE bind multiple 2'OH on the backbone) and the cooperativity of molecules during unwinding (two symmetrical cores of DDX3X bind the RNA together). Alongside the structure, the authors propose a complete unwinding cycle which includes three previously solved structures of the DDX3X core. The structure provides insight into the catalytic cycle of the RNA helicase DDX3X and substrate recognition. The structures are of high quality and the analysis is thorough. There are a few small points that hopefully will be helpful for the authors.

We thank the reviewer for the encouragement.

On page 4 paragraph 2, it is noted that the CTW also interacts with the RNA strand's 2'-OH groups, however, the interaction itself is not shown. Instead, a cartoon schematic indicates that the interaction exists. It would be helpful to see the interaction and not just the schematic, if possible.

We thank the reviewer for the advice. In the revised manuscript, we added Extended Data Fig. 3 to show the detailed interactions between the CTE and RNA.

On page 5, at the end of paragraph 1, there exists a discrepancy between the figure and the text. The text states that "... the closed (post-unwound) and partially closed (apo) states are similar, while the open (pre-unwound) and partially open (apo) states are similar." In figure 3a, the partially open figure is attributed to the AMP bound DDX3X, and not the apo state.

Please elaborate on the partially open state, whether or not AMP binding is used to mimic the apo state and whether AMP binding changes the shape of the molecule from closed to open state.

We thank the reviewer for pointing this out. Representing an apo conformation by the D1D2:AMP structure (PDB: 2I4I)⁵ is an assumption on the basis of the fact that AMP is not relevant to the unwinding reaction. During the revision, we realized that proposing a representative state based on an assumption weakens our conclusions. Therefore, we removed the D1D2:AMP structure together with the proposed partially open state, making the similarities and differences between apo, pre-unwound, and post-unwound states much easier to be visualized and presented (Fig. 5a). Also, we included the D1D2:ADP structure (PDB: 4PXA)⁶ in the functional cycle to represent a post-release state, regarding the release of RNA product in consistent with the pre- and post-unwound nomenclature (Page 9, Line 19 – Page 10, Line 2). It is still a four-state functional cycle, but free from any assumptions (Fig. 5b).

Figure 3 is confusing. It's unclear why these states are necessary as part of the unwinding cycle. Also, it's not clear from the figure where the RNA is located in the closed state.

Fig. 3 of the submitted manuscript is now Fig. 5 in the revised manuscript. As explained above, removing the D1D1:AMP structure (PDB: 2I4I)⁵ together with the partially open state makes the comparison of apo, pre-unwound, and post-unwound states much easier and more informative (Fig. 5a). In the functional cycle, we included the D1D2:ADP structure (PDB: 4PXA)⁶ to represent a post-release state (Fig. 5b). Through transparent molecular surface, the RNA in the closed state can be easily seen.

Page 12 “Data availability. Structure factors and atomic coordinates will be deposited in the Protein Data Bank (PDB) and released upon publication of this manuscript.” This is not good practice – a pdb validation report really must be provided as a condition for publication and would have helped with the review.

We have deposited the D1D2:dsRNA:D1D2 structure of DDX3X in the PDB under the accession code 6O5F.

Minor corrections

Page 4 “Our structure shows that D1 and CTE discriminate DNA from RNA, which was never thought of before.” This should be rephrased – maybe instead write ‘not previously recognised or demonstrated’.

We reworded this statement in the context of Fig. 2d. It reads, “We show here that all four domains (NTE, D1, D2, and CTE) recognize 2'-OH groups and thereby discriminate DNA from RNA substrates (Page 6, Lines 14-16).”

Page 5, 2nd line 2nd paragraph “Although it may not be as ...”

This statement was removed together with the D1D2:AMP structure (PDB: 2I4I)⁵ from the proposed functional cycle.

Page 8, reference 8 incomplete

Reference 8 in the submitted version is Reference 28 in the revised manuscript.

Page 11, methods, half way down the page, need centrifugation in frce, not rpms.

We converted the 15,000 rpm to 26,892 g (Page 12, Line 22).

Figure 3b – typo in label for figure “Partially Open state”

Fig. 3b of the submitted manuscript is Fig. 5b in the revised manuscript. During the revision, we removed the “Partially Open state” together with the D1D2:AMP structure (PDB: 2I4I)⁵ from the proposed functional cycle.

References

- 1 Sharma, D., Putnam, A. A. & Jankowsky, E. Biochemical Differences and Similarities between the DEAD-Box Helicase Orthologs DDX3X and Ded1p. *J. Mol. Biol.* **429**, 3730-3742 (2017).
- 2 Floor, S. N., Condon, K. J., Sharma, D., Jankowsky, E. & Doudna, J. A. Autoinhibitory Interdomain Interactions and Subfamily-specific Extensions Redefine the Catalytic Core of the Human DEAD-box Protein DDX3. *J. Biol. Chem.* **291**, 2412-2421 (2016).
- 3 Putnam, A. A. *et al.* Division of Labor in an Oligomer of the DEAD-Box RNA Helicase Ded1p. *Mol. Cell* **59**, 541-552 (2015).
- 4 Chen, Y. *et al.* DEAD-box proteins can completely separate an RNA duplex using a single ATP. *Proc. Natl. Acad. Sci. U.S.A.* **105**, 20203-20208 (2008).
- 5 Hoggom, M. *et al.* Crystal structure of conserved domains 1 and 2 of the human DEAD-box helicase DDX3X in complex with the mononucleotide AMP. *J. Mol. Biol.* **372**, 150-159 (2007).
- 6 Epling, L. B., Grace, C. R., Lowe, B. R., Partridge, J. F. & Enemark, E. J. Cancer-associated mutants of RNA helicase DDX3X are defective in RNA-stimulated ATP hydrolysis. *J. Mol. Biol.* **427**, 1779-1796 (2015).

Reviewers' comments:

Reviewer #1 (Remarks to the Author):

The revised manuscript by Song and Ji describing the mechanism of RNA duplex recognition and unwinding by DEAD-box helicase DDX3X has been significantly improved due to addition of new experimental results and missing details.

The authors performed a new biochemical experiment indicating that two DDXs function cooperatively and provided a direct evidence for RNA duplex unwinding by two DDXs helicases.

However, there are remaining major concerns:

1.) The previous criticism concerning the possibility of twinning has been discussed in more details. Although the provided explanation sounds plausible, a few more details would be required to eliminate any remaining concerns.

a) The difference in overall Rmerge/Rmeas for a data set processed in P31 and P3121 space groups (also for the highest resolution shell).

b) Gain in crystallographic R-factors between the P3121 and P31 models.

c) Just for curiosity. Did the authors check the orientation of the NCS 2-fold axis in P31 structure? How much does it deviate from the crystallographic 2-fold axis? A very small difference would explain the "false" detection of twinning by PDB validation.

2.)

If I got it right, the conclusion is that the proposed model (two-molecule cooperativity) does actually not apply for full-length DDX3X (three-molecule cooperativity), which of course would be the DDX3X species working in a living cell. So basically this means, that this model does not describe the working mechanism of DDX3X, but potentially of other DDX/DEAD-box helicases lacking the segments omitted in this minimal D1D2 core construct.

This still raises questions that remain unanswered in the manuscript:

- Are there other DDX/DEAD-box helicases known to show the two-molecule cooperativity, for which this model could apply? Are there DDX/DEAD-box helicases that only consist of the minimal components used for this construct? How do they behave? These are important aspects that need to be addressed, as for now the model does not describe the actual working mode of (full-length) DDX3X and thus conclusions like "our unwinding mechanism of DDX3X should be applicable to other DDX proteins" might not be appropriate. Also, in the light of the fact, that it does not really describe the actual DDX3X mechanism, it might be more appropriate to change the title to something more globally applicable like "The mechanism of RNA duplex recognition and unwinding by (minimal?) DEAD-box helicases"

- It is clear how many residues are missing at the N-terminal end by looking at the used residue range, but it is nowhere mentioned how much is missing at the C-terminal end. Does the used construct contain "the low complexity region that is essential for oligomerization" mentioned in the introduction? Could the lack of this region explain the discrepancy in the amount of molecules observed for the different cooperativity scenarios?

- What could be the role of the third molecule in case of full-length DDX3X? Could it also be involved in RNA-binding (which space-wise seems unlikely)?

Minor points:

Page 1 – line 11: the term "non-processive" is usually used in literature

Reviewer #2 (Remarks to the Author):

The authors have provided satisfactory responses to the points raised by both reviewers. The revised manuscript is substantially improved, and there are a few points that require attention:

Regarding crystal twinning, it is not necessarily true that the refinement result proves that the crystal is not twinned. Is there an estimated twin fraction for the data processed as P31 ?

Page 3, Line 17, and throughout text, "Hill kinetics" might be more accurately named "Hill cooperativity coefficient"

Figure 4 legend - note sure that Hill kinetics is the appropriate description of the figure.

Page 3, line 18 "shedding light"

Page 4, line 13 Typo "Our"

Not sure of the logic of the statement on line 17 that the crystal would necessarily exhibit higher symmetry - it could still have the lower symmetry even if the asymmetric unit was symmetric with respect to the RNA:protein contacts (exposed surfaces could differ, for instance).

Page 10 , line 21 "This fact affects mechanistic ..." sentence is very awkward and needs to be re-written

Figure 1 legend, would be helpful to define PDDFs here

Page 13, line 19 "Initially, the data were scaled.."

We thank the reviewers for the encouragement and advice for further improvement. Please see our point-to-point response to the referees' comments below.

Reviewers' comments

Reviewer #1 (Remarks to the Author):

The revised manuscript by Song and Ji describing the mechanism of RNA duplex recognition and unwinding by DEAD-box helicase DDX3X has been significantly improved due to addition of new experimental results and missing details.

The authors performed a new biochemical experiment indicating that two DDXs function cooperatively and provided a direct evidence for RNA duplex unwinding by two DDXs helicases.

However, there are remaining major concerns:

1.) The previous criticism concerning the possibility of twinning has been discussed in more details. Although the provided explanation sounds plausible, a few more details would be required to eliminate any remaining concerns.

a) The difference in overall $R_{\text{merge}}/R_{\text{meas}}$ for a data set processed in $P3_1$ and $P3_121$ space groups (also for the highest resolution shell).

Space group	$P3_1$	$P3_121$
Resolution (Å)	40.00-2.50 (2.59-2.50)*	40.00-2.50 (2.59-2.50)
R_{merge}	0.165 (1.948)	0.187 (2.006)
R_{meas}	0.190 (2.221)	0.202 (2.151)

* In parenthesis are values for the highest resolution shell.

b) Gain in crystallographic R-factors between the $P3_121$ and $P3_1$ models.

Space group	$P3_1$	$P3_121$
RNA sequence	As shown in Extended Data Fig. 2a	As shown in Extended Data Fig. 2b
Resolution (Å)	35.90-2.50	35.90-2.50
$R_{\text{work}}/R_{\text{free}}$	0.213/0.246	0.224/0.276

c) Just for curiosity. Did the authors check the orientation of the NCS 2-fold axis in $P3_1$ structure? How much does it deviate from the crystallographic 2-fold axis? A very small difference would explain the "false" detection of twinning by PDB validation.

We checked the orientation of the NCS 2-fold axis in the $P3_1$ structure during this revision. The deviation of the NCS 2-fold axis in the $P3_1$ structure from the crystallographic 2-fold axis is negligible. As shown on the right, the two axes overlap well. As the reviewer pointed out, "the very small difference explains the "false" detection of twinning by the PDB validation."

2.) If I got it right, the conclusion is that the proposed model (two-molecule cooperativity) does actually not apply for full-length DDX3X (three-molecule cooperativity), which of course would be the DDX3X species working in a living cell. So basically this means, that this model does not describe the working mechanism of DDX3X, but potentially of other DDX/DEAD-box helicases lacking the segments omitted in this minimal D1D2 core construct.

This is a stimulating comment, which encouraged us to think deeper in the following four aspects.

First, the NTE-D1-D2-CTE fragment (residues 132-607) of DDX3X (Fig. 1a), which we refer to as the D1D2 core for the ease of discussion, is the minimal, subfamily-specific functional core of DDX3X (*J. Biol. Chem.* **291**:2412-2421, 2016). Although not as efficient as the full length (residues 1-662), the D1D2 core unwinds dsRNA and hence it is a functional DDX3X protein; in contrast, other fragments (such as residues 132-582 or residues 168-582) completely lost the unwinding activity (*J. Biol. Chem.* **291**:2412-2421, 2016). However, how the D1D2 core unwinds RNA duplex was not known. In this study, we provide strong evidences to demonstrate that two D1D2 cores bind and unwind dsRNA cooperatively.

Second, DDX3X and Ded1p are the closest orthologs; together, they represent a DDX subfamily that features a low complexity region (LCR), located in the C-terminal sequence of ~60 amino acid residues (*Crit. Rev. Biochem. Mol. Biol.* **49**:343-360, 2014). Full-length Ded1p and DDX3X exhibit three-molecule cooperativity as the Hill coefficient ($H \approx 3$) (*J. Mol. Biol.* **429**:3730-3742, 2017). It was previously shown that removing 69 residues from the C-terminus of Ded1p, the truncated DDX could only form dimers instead of forming the trimer (*Mol. Cell* **59**:541-552, 2015). We show in this study that without 55 C-terminal residues, two DDX3X minimal catalytic D1D2 cores exhibit two-molecule cooperativity instead of exhibiting three-molecule cooperativity.

Third, the three-molecule cooperativity model described for full-length Ded1p features a strict division of labor between the three Ded1p protomers (*Mol. Cell* **59**:541-552, 2015). Interestingly, unlike dsRNA-unwinding activity, of which $H \approx 3$, optimal ATP-hydrolyzing activity requires cooperation between two molecules as indicated by $H \approx 2$ (*Mol. Cell* **59**:541-552, 2015). In this study, we show that for both dsRNA-unwinding and ATP-hydrolyzing activities of the DDX3X D1D2 core (residues 132-607), $H \approx 2$. Together, these results suggest that two molecules of DDX3X/Ded1p, with or without the LCR, are sufficient for dsRNA unwinding and ATP-hydrolyzing activity.

Fourth, DDX3X unwind RNA duplex most efficiently as trimers although its D1D2 core also unwinds dsRNA (*J. Biol. Chem.* **291**:2412-2421, 2016; *J. Mol. Biol.* **429**:3730-3742, 2017). Here, we show that two D1D2 cores bind dsRNA and unwind dsRNA with the consumption of two ATP molecules. Together, these observations suggest that the third DDX3X molecule is not essential for function. Instead, it optimizes unwinding efficient without consuming a third ATP molecule.

Taken together, our new structural and functional data on DDX3X are not only in agreement with previous observations, but also significantly further our understanding of the mechanism of dsRNA unwinding by the DDX. We thank the reviewer very much for the stimulating comment and have expanded the Discussion by one more paragraph addressing the discrepancy in the number of molecules for the different cooperativity scenarios (Page 12, Lines 1-12).

This still raises questions that remain unanswered in the manuscript:

- Are there other DDX/DEAD-box helicases known to show the two-molecule cooperativity, for which this model could apply? Are there DDX/DEAD-box helicases that only consist of the minimal components used for this construct? How do they behave? These are important aspects that need to be addressed, as for now the model does not describe the actual working mode of (full-length) DDX3X and thus conclusions like “our unwinding mechanism of DDX3X should be applicable to other DDX proteins”

might not be appropriate. Also, in the light of the fact, that it does not really describe the actual DDX3X mechanism, it might be more appropriate to change the title to something more globally applicable like “The mechanism of RNA duplex recognition and unwinding by (minimal?) DEAD-box helicases”

To the best of our knowledge, there are no other DDX/DEAD-box helicases that only consist of the NTE-D1-D2-CTE fragment (residues 132-607) of DDX3X.

We have replaced the statement “our unwinding mechanism of DDX3X should be applicable to other DDX proteins” with “our unwinding mechanism of DDX3X provides a new model for mechanistic studies of other DDX proteins (Page 12, Lines 14-15).”

As mentioned above, our new structural and functional data are not only in agreement with previous observations, but also significantly further our understanding of the mechanism of dsRNA recognition and unwinding by DDX3X. Therefore, the title is still valid, and we would like to keep it unchanged.

- It is clear how many residues are missing at the N-terminal end by looking at the used residue range, but it is nowhere mentioned how much is missing at the C-terminal end. Does the used construct contain “the low complexity region that is essential for oligomerization” mentioned in the introduction? Could the lack of this region explain the discrepancy in the amount of molecules observed for the different cooperativity scenarios?

We thank the reviewer very much for pointing this out. Yes, it is the lack of the low complexity region that explains the discrepancy in the number of molecules for the different cooperativity scenarios. The information about the low complexity region and how it explains the discrepancy in the number of molecules for the different cooperativity scenarios has been added (Page 12, Lines 1-12).

- What could be the role of the third molecule in case of full-length DDX3X? Could it also be involved in RNA-binding (which space-wise seems unlikely)?

The three-molecule cooperativity described for Ded1p features a division of labor (RNA-loading and RNA-unwinding) between the three protomers. We believe that this is also true for DDX3X, suggesting that the third protomer plays the RNA-loading role and thereby optimizes the RNA-unwinding efficiency. It is clear the low complexity region is required for the recruitment of the third protomer, but how the third protomer associates with the other two via the low complexity region is not clear (Page 12, Lines 6-12).

Minor points:

Page 1 – line 11: the term “non-processive” is usually used in literature

Corrected as advised (Page 1, Line 11).

Reviewer #2 (Remarks to the Author):

The authors have provided satisfactory responses to the points raised by both reviewers. The revised manuscript is substantially improved, and there are a few points that require attention:

Regarding crystal twinning, it is not necessarily true that the refinement result proves that the crystal is not twinned. Is there an estimated twin fraction for the data processed as P31 ?

We agree with the reviewer. Twinning can be ruled out without the refinement.

The estimated twin fraction for (h, -h-k, -l) is 47.0% for the data processed as P3₁.

Page 3, Line 17, and throughout text, "Hill kinetics" might be more accurately named "Hill cooperativity coefficient"

We have changed the "Hill kinetics" to "Hill cooperativity analysis."

Figure 4 legend - note sure that Hill kinetics is the appropriate description of the figure.

We have changed legend to "Hill cooperativity analysis of the ATPase activity of human DDX3X (Page 25, Line 16)."

Page 3, line 18 "shedding light"

Corrected as advised (Page 3, Line 18).

Page 4, line 13 Typo "Our"

We have corrected this typo (Page 4, Line 13).

Not sure of the logic of the statement on line 17 that the crystal would necessarily exhibit higher symmetry - it could still have the lower symmetry even if the asymmetric unit was symmetric with respect to the RNA:protein contacts (exposed surfaces could differ, for instance).

We thank the reviewer for pointing this out and understand that the crystal may still have the lower symmetry even if the two protein:ssRNA complexes are identical. Accordingly, we have revised the sentence, using the clause "the crystal could exhibit higher symmetry (Page 4, Lines 15-17)."

Page 10 , line 21 "This fact affects mechanistic ..." sentence is very awkward and needs to be re-written

We rewrite this sentence "This observation indicates that dsRNA recognition and unwinding mechanism is different from that previously proposed based on the D2:dsRNA structure (Page 10, Lines 18-20)."

Figure 1 legend, would be helpful to define PDDFs here

We have defined PDDF in the legend of Figure 1 (Page 24, Lines 11-12).

Page 13, line 19 "Initially, the data were scaled.."

We have corrected this error (Page 14, Line 10).

REVIEWERS' COMMENTS:

Reviewer #1 (Remarks to the Author):

The authors addressed all comments, hence, this revised version of manuscript should be accepted as it is.